# Association between Individual Norepinephrine Transporter (NET) Availability and Response to Pharmacological Therapy in Adults with Attention-Deficit/Hyperactivity Disorder (ADHD)

**DOI:** 10.3390/brainsci12080965

**Published:** 2022-07-22

**Authors:** Jue Huang, Nicole Mauche, Michael Rullmann, Christine Ulke, Georg-Alexander Becker, Marianne Patt, Franziska Zientek, Swen Hesse, Osama Sabri, Maria Strauß

**Affiliations:** 1Department of Psychiatry and Psychotherapy, University of Leipzig Medical Center, 04103 Leipzig, Germany; nicole.mauche@medizin.uni-leipzig.de (N.M.); christine.ulke@medizin.uni-leipzig.de (C.U.); maria.strauss@medizin.uni-leipzig.de (M.S.); 2Department of Nuclear Medicine, University of Leipzig Medical Center, 04103 Leipzig, Germany; michael.rullmann@medizin.uni-leipzig.de (M.R.); georg.becker@medizin.uni-leipzig.de (G.-A.B.); marianne.patt@medizin.uni-leipzig.de (M.P.); franziska.zientek@medizin.uni-leipzig.de (F.Z.); swen.hesse@medizin.uni-leipzig.de (S.H.); osama.sabri@medizin.uni-leipzig.de (O.S.)

**Keywords:** ADHD, psychostimulant, non-stimulant, comorbidity, depression, anxiety, follow-up, therapy response

## Abstract

Background: The role of the norepinephrine transporter (NET) has received increased focus in recent studies on the pathogenesis of attention-deficit/hyperactivity disorder (ADHD). The predictive value for pharmacological treatment and its link to other health or social limitations has been little-studied. This follow-up research on adult patients with ADHD aimed to explore whether the therapy response and health and social impairments depend on baseline individual NET availability. Methods: Data were collected from 10 patients on personal, family and professional situations, mental and physical health and treatments received after baseline via online and telephone surveys and were compared to baseline data to evaluate treatment-related changes. Results: The majority of our ADHD patients did not show therapy responses but showed improvements due to pharmacological treatment. There was no evidence of relationships between pre-treatment NET availability and therapy response or health/social limitations. Conclusions: Pharmacological monotherapy was insufficient to promote symptom remission, especially for participants with extreme insufficiency in NET availability, but improved outcomes in academic and social functioning. Psychotherapy should be considered as an add-on to the standard treatment approach due to its positive outcome in reducing social limitations. The prognostic value of individual NET availability in predicting the response to therapy needs further studies with large sample sizes.

## 1. Introduction

Attention-deficit/hyperactivity disorder (ADHD) is a childhood-onset neurodevelopmental disorder characterized by core symptoms of inattention, hyperactivity and impulsivity [1]. Its symptoms in about 60% to 80% of pediatric patients persist into adulthood [2,3] and result in profound impacts on academic performance, social interaction and financial functioning [4]. Recent studies show increased comorbidity between ADHD and other psychiatric disorders such as mood disorders, substance abuse and anxiety.

The causes of ADHD are complex, and the underlying etiology and pathophysiology remain unclear. A growing body of evidence support a multifactorial model in which a dopamine (DA) and/or norepinephrine (NE) deficiency in several brain regions is an important risk factor of this disorder [5,6]. In our prior study, unmedicated adults suffering from ADHD yielded decreased NE transporter (NET) availability in brain regions relevant for attention, which indicated the pathophysiological involvement of NET availability in adult ADHD [7]. These findings were perfectly in accordance with the mechanism of action of the medication used to treat ADHD in adults according to the treatment guideline [8].

The exact mechanism of action of ADHD medication is still unclear. The generally believed mechanism of action is the effect of a psychostimulant (e.g., methylphenidate (MPH) or lisdexamfetamine (LDX)) and a non-stimulant (i.e., atomoxetine (ATX)) on the DA system due to their high affinity for the DA transporter (DAT) [9,10,11]. However, the high affinity of MPH [12,13] and ATX [14,15] for NET both in vitro and in vivo has recently been evidenced, and MPH showed an even higher affinity for NET than DAT [13]. With considerable evidence for the effectiveness of both psychostimulants (i.e., MPH) and non-stimulants (i.e., ATX) in humans with ADHD, several findings suggested that the therapeutic effect of the medication used to treat ADHD might be mediated by NET availability [16,17,18]. This issue, however, has not been investigated.

The efficiency and safety of the medication for ADHD was intensively examined. Previous findings suggest that the individual response to medication can vary significantly. About 30% to 42% patients cannot benefit from their medication [19,20]. In the most recent clinical trial, more than 90% of patients showed symptomatology improvement, but only 30% of them reached the defined remission [21]. Explanations for the varied responses include personal clinical characteristics, gene type, symptom severity, dosage, therapy duration and co-occurrence of comorbidities. The mentioned research evidence suggests that there is a need to take individual characteristics that might be associated with the response to treatment for ADHD into account. In light of the above-mentioned evidence, this study mainly aimed to explore whether the response to treatment was related to individual initial NET availability.

Additionally, recent studies show increased comorbidity between ADHD and other psychiatric disorders such as mood disorders, substance abuse and anxiety. With a consideration of the modulating role of NET in autonomic and cognitive functions such as arousal, attention and mood [22], we aimed to explore whether the possible health or, consequently, social limitations are related to individual initial NET availability.

In short, this study aimed to explore the possible answers to the questions of whether (1) the response to treatment and (2) the possible health-related or social limitations are related to individual initial NET availability. To explore our hypotheses in this study, we planned a follow-up study to collect data of participants from our prior study [7] about personal, family and professional situations and mental and physical health as well as any treatments that have taken place since the previous study. These data have been related to the baseline data to assess their individual responses to treatment.

## 2. Materials and Methods

### 2.1. Participants

For the current study, we invited all recruited participants from the prior study [7] with a diagnosis of ADHD for this follow-up research. The participants were only included for further analysis if they completed online and telephone surveys. There were no further exclusion criteria. The declaration of consent to this study and the consent to the collection and processing of personal data was given as a part of the online survey. All participants gave consent before they started the online survey. This study was approved by the local ethics committee of the University of Leipzig (155/15-ek).

### 2.2. Design and Measurements

In order to explore our hypotheses, the existing baseline test battery was adapted. In this study, participants were only provided with the change-sensitive questionnaires as an online survey with a personalized assess code by email. The following questionnaires were used in the online survey: The German version of *Conners’ Adult ADHD Rating Scale* (CAARS) [23] provided a comprehensive assessment of the presence and severity of ADHD symptomatology with multiple subscales. The German *Beck Depression Inventory-II* (BDI-II) [24] was applied to assess the subjective severity of depressive symptomatology during the last two weeks. The self-reported *Alcohol Use Disorder Identification Test* (AUDIT) [25] and *Drug Use Disorder Identification Test* (DUDIT) [26] served as evaluations of alcohol and/or drug abuses. Criterion C of the semi-structured Diagnostic Interview for ADHD in adults (DIVA-C) [27] was used to evaluate limitations related to home, interpersonal and occupational activities. The *German version of the Patient healthy Questionnaire* (PHQ-D) [28] was supplemented to the baseline test battery, enabling the quick identification of most common mental disorders.

After the online survey, a telephone interview was carried out in which possible changes in sociodemographic data and medical anamnesis were able to be queried. Moreover, the German version of the *Montgomery–Asberg Depression Rating Scale* (MADRS) [29] was applied to assess the objective severity of depressive symptomatology during the last week.

### 2.3. Initial NET Availability

In our prior study, diagnosed unmedicated ADHD adults were examined to test the a priori hypothesis that central NET availability is altered in adult ADHD compared to healthy controls [7]. The baseline assessment of the NET availability in the prior study was specified as the initial NET availability in this study. The values were quantified as a regional distribution of ratios (DVR). The assessment of NET availability in a related region of interest (ROI) was described in detail elsewhere [7].

### 2.4. Definition of Therapy Response, Health Limitations and Social Limitations

In this study, a therapy response was based on a CAARS concept of symptomatic remission defined by a T-score in the CAARS DSM-Global subscale below 60 [23]. Furthermore, for exploratory purposes, we also considered participants showing overall improvements in CAARS DSM-Global total raw scores (i.e., the sum of all item scores in this subscale) under the consideration that the commonly utilized CAARS T-scores are probably change-insensitive for those participants with profound symptoms or those showing minimal improvement.

A health limitation was defined as the existence of diagnosed psychiatric, psychological or physical diseases.

The numbers of impaired aspects of each setting of social life according to the DIVA-C were added to produce an impairment score in each setting. The level of social limitations was indicated by the resultant z-scores: no apparent impairments (z > 1.0), low impairments (0 < z < 1), moderate impairments (−1.0 < z < 0) and high impairments (z < −1.0).

### 2.5. Statistical Analysis

Due to differences in individual symptoms, treatment options or durations, we briefly describe each participant’s case and summarize several common characteristics from these cases.

In this study, comparisons of means between different groups were executed, despite the small sample size. However, the regular sample *t*-tests are unacceptable in the case of unequal sample sizes and when larger variances come from small samples because these tests could provide high false-positive rates or very low statistical power [30]. If these cases occur, we can only provide descriptive statistics; otherwise, regular sample *t*-tests were planned. Detailed results are summarized in the Appendix A.

To minimize the number of comparisons and correlation analyses, we summarized the ROIs of mean DVR into two relevant brain regions based on the finding of a prior study [7]. They are attention- (including the superior frontal gyrus, precuneus, angular and supramarginal gyri, cerebellum with crus and thalamus) and behavioral-control-related ROIs (including the inferior frontal gyrus, anterior cingulate, supplementary motor area, nucleus caudate, putamen and pallidum) for the left and right hemisphere, respectively. Exploratory post hoc analyses for each ROI were planned in the case of significant results. The correlational (Pearson’s r) analyses were conducted separately for the attention- and behavior-related ROIs in both the left and right hemispheres. The adjusted alpha level of 0.0125 per test was applied to avoid type I error inflation. This adjusted alpha level was calculated by the number of tests (n = 4, attention- and behavior-related, for left and right hemispheres). All analyses were performed using the software SPSS Statistics 27.0 (IBM Corp.; Armonk, NY, USA).

## 3. Results

### 3.1. Description of Sample

The final sample involved 10 adults suffering from ADHD who completed both the online and telephone surveys. An overview about their demographic and related clinical profiles prior to the study and at follow-up are provided in Table 1. We briefly describe each participant’s case and summarize their common characteristics.

#### 3.1.1. Case Description

ID 01

A 26-year-old male participant showed, after MPH treatment at a dose of 80 mg/day for about 26 months, no overall improvement in ADHD symptoms at follow-up. There were no other comorbidities, and no related treatments were requested.

ID 02

A 26-year-old female participant showed, after MPH treatment at a dose of 20 mg/day for about 21 months, a slight overall worsening in ADHD symptoms. She reported, in the telephone survey, diagnosed comorbid depressive and anxiety symptoms, consistent with the results from BDI, MADRS and PHQ-D. A multimodal approach that combined psychotherapy and pharmacotherapy was required due to comorbidities.

ID 03

A 53-year-old male participant showed, after MPH treatment at a dose of 40 mg/day for about 8 months, a minimal improvement in ADHD symptoms. There were no other comorbidities, and no related treatments were requested.

ID 04

A 35-year-old male participant showed, after MPH treatment at a dose of 40 mg/day for about 39 months, an obvious overall improvement in ADHD symptoms, which even reached symptomatic remission as defined by Christiansen et al., 2014. Furthermore, he reported a marked improvement in comorbid depressive symptoms, as determined by the BDI and MADRS, even in the absence of a confirmed diagnosis of depression. No reports about other comorbidities or treatments.

ID 05

A 36-year-old female participant showed, after MPH treatment at a dose of 20 mg/day for about 37 months, a minimal worsening of ADHD symptoms. She was recognized to have some comorbid depressive symptoms/syndrome and substance use disorders within last 12 months based on the scores of the BDI, PHQ-D and AUDIT/DUDIT, respectively. However, no related treatments were considered.

ID 06

A 39-year-old female participant showed, after a pharmacological combination of ATX and LDX in respective doses of 40 mg and 70 mg per day for about 22 months, a slight improvement in ADHD symptoms. The reported comorbid depressive symptoms were in line with the results from the BDI, MADRS and PHQ-D. For this comorbidity, she received psychotherapy combined with TMS.

ID 07

A 26-year-old male participant showed, after an irregular MPH intake at a dose of 10 mg/day with parallel sport and neurofeedback, a slight improvement in ADHD symptoms. Comorbid depressive and anxiety symptoms were present since the prior study; however, no real improvement was reported after a psychotherapy.

ID 08

A 34-year-old male participant showed, after MPH treatment at a dose of 10 mg/day for about 18 months, a worsening of ADHD symptoms. There was an indication of some depressive symptoms according to MADRS, but this was inconsistent with the BDI and PHQ-D. There were no other comorbidities, and no related treatments were required.

ID 09

A 34-year-old male participant showed, after a pharmacological combination of MPH at a dose of 30 mg/day, ATX at a dose of 80 mg/day and LDX at a dose of 70 mg/day for about 30 months, a marked improvement in ADHD symptoms, which almost reached symptomatic remission based on Christiansen et al., 2014. He reported some depressive symptoms based on the results from both the BDI and MADRS in the absence of a confirmed diagnosis. No reports about other comorbidities or treatments.

ID 10

A 38-year-old female participant showed, after MPH treatment at a dose of 60 mg/day for about 40 months, an obvious improvement in ADHD symptoms. Furthermore, she was reported to have some comorbid depressive symptoms, which were clearly reduced after psychotherapy. There were no other comorbidities, and no related treatments were required.

#### 3.1.2. Case Summary

In the current study, all 10 participants with ADHD were administered at least one psychostimulant as their treatment for ADHD, and 90% of these cases included MPH. The rate of participants with the administration of a stimulant, i.e., MPH, as a monotherapy was 70%. An additional 20% of the participants were administered a combination of stimulant(s) and ATX. For the remaining 10%, non-psychopharmacological interventions (i.e., sport and neurofeedback) were further required in the case of the irregular intake of MPH.

There was no participant who preferred psychotherapy as a primary or co-treatment for ADHD. A total of 40% of participants required additional psychotherapy due to comorbid depression and/or an anxiety disorder diagnosed by their own psychiatrist or psychologist but not due to ADHD.

The rate of participants reporting an overall improvement in CAARS DSM-Global total raw scores was 60%, but only 10% of these cases matched the definition of a therapy response with a T-score on the CAARS DSM-Global subscale below 60.

Accordingly, all participants with diagnosed ADHD experienced some or considerable limitations: there were five participants classified as having low impairments, four with moderate impairments and one with significant impairments in different settings of life. It was worth noting that all participants who received psychotherapies due to diagnosed comorbidities reported overall less impairments compared to participants without comorbidities/physical disease. Interestingly, the only participant experiencing significant limitations was the one that presented remission based on the CAARS DSM-Global subscale.

### 3.2. Association of NET Availability with Health or Social Limitations and Improvement Due to Treatment at the Group Level

Regarding the group difference in DVRs between participants with and without health limitations, independent-sample t-tests were, in this case, acceptable since the variances in the larger sample were higher than in the smaller sample. No significant group difference was found (0.52 ≤ *p* ≤ 0.96).

Regarding the group difference in DVRs between participants with different levels of social limitations, no sample t-tests were performed due to the extremely unequal and small sample sizes.

Regarding the association between DVRs and health or social limitations, correlational analyses revealed no significant relationships between DVRs and the existence of health limitations (0.52 ≤ *p* ≤ 0.96) or the levels of social limitations (0.72 ≤ *p* ≤ 0.87).

The detailed results of the statistical correlational analyses and the comparisons between the different groups are summarized in the Appendix A.

### 3.3. Association of NET Availability with Health or Social Limitations and Improvement Due to Treatment at the Individual Level

The averaged mean DVRs of NET availability over selected ROIs for each participant are illustrated in Figure 1. In total, 6 out of 10 participants showed some improvement in the CAARS DSM-Global raw scores and are indicated as red in Figure 1. Those participants without any improvements are indicated as black. Overall, the mean DVRs distributed discretely, However, for those without improvements in ADHD symptoms, a trend showed extreme decreased mean DVRs at all ROIs, e.g., ID 05, could be observed. Moreover, a balance in the mean DVRs between the left and right ROIs (i.e., the mean DVR in the left and right ROI at a similar level) for ID 04, the only one that reported a real therapy response, could be seen. Conversely, participant ID 02, who reported multiple comorbidities, showed an imbalance in the mean DVRs between the left and right ROIs, in particular in attention-related areas.

In line with the results for correlational analyses, we could not demonstrate any clear trends for a linear association between individual mean DVRs and social or health limitations. Participants who received psychotherapy for their comorbidities are indicated by filled signs in Figure 1. There was only a trend towards experiencing fewer social limitations when they considered psychotherapies for their comorbidities and showed some improvements in ADHD symptoms based on the DSM-Global raw scores.

## 4. Discussion

Overall, the results of the present study aiming to explore the association between NET availability and therapy response as well as health- or social-related limitations show that all our participants were administered at least one medication to treat ADHD, primarily psychostimulants. The rate of therapy response was very low, even after a long period of medication. There are many different reasons. Depression and anxiety disorders are two comorbidities that were frequently reported by our participants. There was no evidence for clear relationships between NET availability and therapy response or health/social limitations due to a low response rate and the small sample. However, based on these results, some aspects of concern about the diagnosis, treatment or management of ADHD in adults have emerged.

First, the T-score in the CAARS DSM-Global subscale [23] is probably less adequate as a measurement of treatment response. In a previous study [21], 22.3% of ADHD patients reached very high CAARS values that were all transformed to 90 as T-scores. In this study, this percentage reached 40%. It can be speculated that CAARS T-scores may not have adequately captured the effects of treatment over time and thus responders were mistakenly labelled as non-responders.

One fact of this study that cannot be overlooked is that most ADHD patients suffered from at least one significant burden in their life. A total of 60% of our participants did not graduate, and 40% of them were unemployed prior to the study. These rates dropped to 20% and 10%, respectively, at this follow-up study. We speculate that there were some positive outcomes from the received treatments on the academic domain and social functioning. This result is, to some degree, in line with a previous long-term study in patients with ADHD that showed that treated ADHD, although not usually to a normal level, improved long-term outcomes in all categories, e.g., academic, the co-occurrence of psychiatric disorders, self-concept and social functions, compared to untreated ADHD [31]. In this study, the pharmacological therapy, whether psychostimulants or non-stimulants, seemed to reduce the majority of our participants’ academic and social burdens and thereby enhanced their quality of life.

However, four participants (ID 02, ID 06, ID 07, and ID 10) reported developing depressive disorders since the prior study [7]. One of them (ID 07) further reported an anxiety disorder. Another one (ID 02), in addition to an anxiety disorder, also suffered from a sleep disorder. Consistent with the previous findings, ADHD patients had a higher rate of developing comorbidities [2,32]. These two participants, especially the one with multiple disorders, showed severe ADHD symptoms in the prior study and few changes upon treatment. There was evidence showing that anxiety disorders, as a group, influenced the persistence of ADHD [2]. This result, however, could not been replicated by a later 7-year follow-up study [33]. Whether the co-occurrence of the anxiety disorder reported by our participants influences the persistence of ADHD could not be clearly answered. However, it would be intuitive to make an affirmative answer when considering that ADHD and anxiety require different treatment approaches. Without any empirical evidence, this knowledge should be considered when selecting treatment approaches for these patients.

Furthermore, there was one participant (ID 02) who received citalopram due to diagnosed depression (see Table 1). Citalopram seems to increase the activity of DAT, which could possibly be detrimental to ADHD. One of the neurobiological approaches underlying the pathophysiology suggested that ADHD could be a result of a decreased level of DA in the striatum, which is probably caused by too much DAT. The pre-synaptic released DA is absorbed by DAT from the synaptic gap before it can dock in the post-synaptic domain. Therefore, too much DAT could aggravate ADHD symptoms. Along these lines, we did observe a worsening of ADHD symptoms based on the CAARS subscale scores. The therapy response of such patients might be influenced either alone by the co-occurring depression and/or anxiety disorder or by the interaction with the medication administered for this comorbid disorder.

It should be noted that participants with co-occurring comorbidities reported low levels of social limitations compared to those without any comorbidities. We assume that this may be the result of the psychotherapy they received. Psychotherapy, through promoting recovery from depressive disorder, also contributed to an improvement in the quality of life, probably by reducing various impairments in social domains. Although none of the participants in this study considered psychotherapy and the evidence regarding the efficiency and effectiveness of psychotherapy in the sample suffering from ADHD are still preliminary [34], it should be considered as an add-on therapy to the standard pharmacological treatment. This approach requires further studies before being integrated into clinical practice.

Failure to demonstrate any associations of NET availability could probably be explained by the large variance due to the small sample size. However, the observation that participants with decreased NET availability tended to show little improvement after treatment might indicate that monotherapy with MPH is not enough for ADHD patients with extremely insufficient NET availability. Other treatment approaches, e.g., stimulation therapy, which directly regulates the activity of neurons at specified brain regions during a stimulation session, could be considered as supplementary therapies for these patients. Additionally, the observation about a potential relationship between NET availability hemisphere imbalances and co-occurred depressive/anxiety disorder might imply that this imbalance could play a role in the presence of mood and anxiety disorders. This issue should be studied further and replicated in a large sample.

Some limitations of this study should be mentioned. The main limitation was the inability to generalize our results and the risk of explaining error due to the extremely small sample size. A second limitation was the lack of classification of ADHD subtypes in the prior study, which provided no related information for this follow-up research. Furthermore, the online survey and telephone interview could be seen as methodological limitations due to the lack of intuitive clinical observations. On the other hand, the main contribution of study should also be mentioned, namely, the prominent importance of the individual aspect in the treatment or management of this disorder, and the importance of studying the predictive value of NET availability was highlighted.

## 5. Conclusions

Although the majority of adult participants with ADHD did not respond to the pharmacological treatment according to our definition of response, they seemed to benefit from the regular treatment by showing reduced burden in the academic and social domains. The issue preventing the effect of pharmacological treatment from reducing comorbidities should be investigated in further studies with sufficient sample sizes. Moreover, some improvements were still observable in ADHD symptoms. A monotherapy with MPH might be not adequate for those patients with extremely insufficient NET availability. A multimodal therapy consisting of ADHD-specific medication and psychotherapy, as recommended by the therapy guidelines for adult ADHD patients, seems to be the better therapeutic option and led, in our study, to a reduction in social limitations and improved the patients’ quality of life. Further studies with proper sample sizes are needed to clarify issues related to the association between NET availability and the therapy response.

## Figures and Tables

**Figure 1 brainsci-12-00965-f001:**
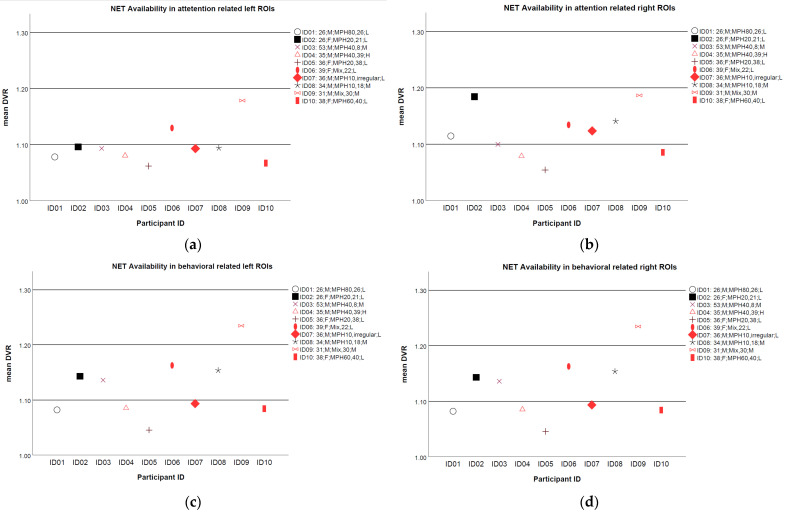
Mean distribution volume ratios (DVRs) of NET availability for each participant in (**a**) attention-related left selected regions of interest (ROIs); (**b**) attention-related right ROIs; (**c**) behavior-related left ROIs; and (**d**) behavior-related right ROIs. Participants with some improvements in CAARS DSM-Global raw scores are marked as red. Participants with diagnosed comorbidities are indicated by filled signs. Note: description in the legend represents, in turn, participant ID, age, sex, medication name, duration in months and level of social limitations. L, low limitations; M; moderate limitation level; H, significant limitations.

**Table 1 brainsci-12-00965-t001:** Demographic and related clinical profiles of adults with ADHD prior to the study and at follow-up.

ID	01	02	03	04	05	06	07	08	09	10	Mean	SD
Age (years) ^a^	26	26	53	35	36	39	26	34	31	38	34.3	8.2
Sex	M	F	M	M	F	F	M	M	M	F		
Therapy response ^1^	N	N	Y	Y	N	Y	Y	N	Y	Y		
Medical therapy dose (mg)	MPH 80	MPH 20	MPH 40	MPH 40	MPH 20	ATX 40 LDX 70	MPH 10	MPH 10	MPH 30 ATX 80 LDX 70	MPH 60		
Duration (months) ^b^	26	21	8	39	38	22	/ ^2^	18	30	40		
Other therapy	None	None	None	None	None	None	ST NFB	None	None	None		
Complications	None	MDD AD SD	None	None	None	MDD	MDD AD	None	None	MDD		
Therapy for complication	None	PSYT CITA 40, 23 m	None	None	None	PSYT TMS	PSYT	None	None	PSYT		
Physical diseases	None	None	None	None	None	None	None	None	None	LTHY		
Marital status												
Prior	S	S	P	P	S	S	S	P	S	P		
Follow-up	P	S	P	P	P	S	S	P	D	P		
Graduation												
Prior	N	N	N	Y	Y	N	N	Y	N	Y		
Follow-up	Y	Y	N	Y	Y	Y	N	Y	Y	Y		
Job status												
Prior	STU	UE	E	UE	E	E	STU	E	E	E		
Follow-up	E	E	E	E	E	E	STU	E	E	E		
CAARS DSM-Global (R)												
Prior	33	42	32	36	28	42	44	24	39	45	36.5	7.1
Follow-up	34	49	31	8	30	38	41	32	18	25	30.6	11.6
CAARS DSM-Global (T)												
Prior	76	90	90	84	71	89	90	68	84	90	83.2	8.5
Follow-up	77	90	88	46	74	84	86	79	60	67	75.1	13.9
BDI												
Prior	3	7	4	11	10	0	33	1	17	40	12.6	13.7
Follow-up	0	36	3	5	17	19	45	2	10	8	14.5	15.2
MADRS												
Prior	11	7	7	21	15	8	19	7	6	15	11.6	5.5
Follow-up	2	24	7	0	12	23	27	15	10	4	9.2	7.9
AUDIT												
Prior	3	1	1	5	4	4	12	6	7	3	4.6	3.2
Follow-up	2	1	1	4	8	3	6	4	4	1	3.4	2.3
DUDIT												
Prior	0	0	0	4	3	0	0	8	5	0	2.0	2.9
Follow-up	0	0	0	0	7	0	2	5	0	0	1.4	2.5
DIVA-C												
Prior	n.a.	n.a.	n.a.	n.a.	n.a.	n.a.	n.a.	n.a.	n.a.	n.a.		
Follow-up												
Work/Education	L	M	M	M	None	M	H	M	M	None		
Relationship	M	M	M	M	M	None	None	M	L	L		
Social contacts	L	None	L	H	M	L	L	H	H	None		
Hobby	M	L	M	H	L	M	None	L	M	L		
Self-confidence	H	None	None	H	M	M	None	M	M	None		
Overall	M	L	M	H	L	L	L	M	M	L		
PHQ-D												
Prior	n.a.	n.a.	n.a.	n.a.	n.a.	n.a.	n.a.	n.a.	n.a.	n.a.		
Follow-up												
Depressive syndrome	3	22	5	5	11	15	25	3	9	8	10.6	7.8
Panic syndrome	N	N	N	N	N	N	Y	N	N	N		
Other anxiety syndrome	0	14	1	1	1	12	14	1	0	1	4.5	6.1
Somatic syndrome	1	17	1	9	14	9	16	1	2	12	8.2	6.5
Psychosocial stressors	1	12	0	7	8	9	18	1	5	8	6.9	5.5
Bulimia nervosa	N	N	N	N	N	N	N	N	N	N		
Binge-eating disorder	N	N	N	N	N	N	Y	N	N	N		
Alcohol syndrome	N	N	N	N	Y	N	N	N	N	N		

^a^ Age in years at follow-up; ^b^ Duration was estimated after the prior research; ^1^ therapy response referred to the overall improvement in CAARS DSM-Global total raw scores; there was only one patient could be treated as a responder based on the definition according to Christiansen et al., 2014; ^2^ irregular medication intake; Note: SD, standard deviation; M, male; F, female; Y, yes; N, no; MPH, methylphenidate; ATX, atomoxetine; LDX, lisdexamfetamine; ST, sport therapy; NFB, neurofeedback; MDD, major depressive disorder; AD, anxiety disorder; SD, sleep disorder; PSYT, psychotherapy; CITA, citalopram; LTHY, hypothyreose; m, month; TMS, transcranial magnetic stimulation; S, single; P, partnership; D, divorced; STU, student; E, employed; UE, unemployed; L, low impairment; M, moderate impairment; H, significant impairment; n.a., not available.

## Data Availability

The dataset supporting the conclusions of this article is included within the article. The spreadsheets and corresponding syntax are available on request from J.H. M.R. had access to all the PET-MRI data and takes responsibility for the integrity and accuracy of the modeling of the PET-MRI data.

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
