# Peer review of "Association between Individual Norepinephrine Transporter (NET) Availability and Response to Pharmacological Therapy in Adults with Attention-Deficit/Hyperactivity Disorder (ADHD)"

_brainsci, 2022, doi:10.3390/brainsci12080965_

Round 1
Reviewer 1 Report
Huang et al reported a follow-up research and released the data collected from 10 ADHD patients treated with methylphenidate (MPH). They found that pharmacological monotherapy was insufficient to promote symptom remission. Although the sample size is small, the data are valuable to the clinical treatment for ADHD with MPH.
Please detail the treatment for the patients. For example, “ID 01, after 80mg MPH monotherapy for about 26 months”, does this mean the patient took only one dose of MPH (80 mg), and did the analysis 26 months later or took MPH 80 mg daily for 26 months? If the patients only take one dose of MHP, do they have acute effect within hours or several days after the MHP dose?
Add a blank between number and unit, like line 167, 80mg to 80 mg.
Reviewer 2 Report
This study is an important study which has addressed ADHD as a complex issue among the adult population. It is appreciated that ADHD was frame worked out from the traditional framework.
The study used SPSS as a quantitative data analysis. But it is not comprehensively explained the purpose of using it. The present study could have been clearly focused on Research questions, Hypothesis followed by and the sample items of the questionnaire used in order to portraying a realistic image of the quantitative approach of the research.
Round 2
Reviewer 1 Report
The author has responded my questions.